# END-TO-END MULTI-LINGUAL MULTI-SPEAKER SPEECH RECOGNITION

## ABSTRACT

The expressive power of end-to-end automatic speech recognition (ASR) systems enables direct estimation of the character or word label sequence from a sequence of acoustic features. Direct optimization of the whole system is advantageous because it not only eliminates the internal linkage necessary for hybrid systems, but also extends the scope of potential application use cases by training the model for multiple objectives. Several multi-lingual ASR systems were recently proposed based on a monolithic neural network architecture without language-dependent modules, showing that modeling of multiple languages is well within the capabilities of an end-to-end framework. There has also been growing interest in multi-speaker speech recognition, which enables generation of multiple label sequences from single-channel mixed speech. In particular, a multi-speaker end-to-end ASR system that can directly model one-to-many mappings without additional auxiliary clues was recently proposed. In this paper, we propose an all-in-one end-to-end multi-lingual multi-speaker ASR system that integrates the capabilities of these two systems. The proposed model is evaluated using mixtures of two speakers generated by using 10 languages, including mixed-language utterances.

## 1 INTRODUCTION

The expressive power of an end-to-end automatic speech recognition (ASR) system enables direct conversion from input speech feature sequences to output label sequences without any explicit intermediate representations and hand-crafted modules (Amodei et al., 2015; Soltau et al., 2016; Hori et al., 2017; Chorowski & Jaitly, 2016; Chan et al., 2016). In addition to eliminating these intermediate linkage components found in hybrid systems, the direct optimization of the whole end-to-end system allows the model to be more easily targeted to different scenarios simply by changing the training data and objectives.

Multi-lingual speech recognition is one such scenario, in which the goal is to support recognition of multiple languages. Conventional approaches require language dependent modules and rely on a pipeline processing consisting of language identification followed by recognition of speech with matched language-dependent system. However, recent studies have demonstrated end-to-end systems that can recognize multiple languages without language dependent modules (Watanabe et al., 2017a; Seki et al., 2018b; Toshniwal et al., 2018). These methods eliminate the need for a language identification module, making it easier for an application developer to produce systems for an arbitrary set of languages.

Whereas conventional ASR systems support recognition of speech by a single speaker, it is typically difficult or impossible to use them in scenarios where multiple people are talking simultaneously. There has recently been growing interest in dealing with such situations, with many developments in the field of single-channel multi-speaker ASR (Cooke et al., 2009; Rennie et al., 2010; Hershey et al., 2010; 2016; Isik et al., 2016; Chen et al., 2017; Qian et al., 2017; Yu et al., 2017; Chen et al., 2018). The goal of single-channel multi-speaker speech recognition is to recognize the speech of multiple speakers given the single-channel mixture of their acoustic signals, in a one to many transformation. Promising techniques have been proposed for this task, but earlier works have required the availability of additional training information such as the isolated source signals of each speaker (Isik et al., 2016) or the phonetic state alignments (Qian et al., 2017; Yu et al., 2017) for effective learning. Some of these also require an explicit intermediate separation stage prior to recognition (Isik et al.,

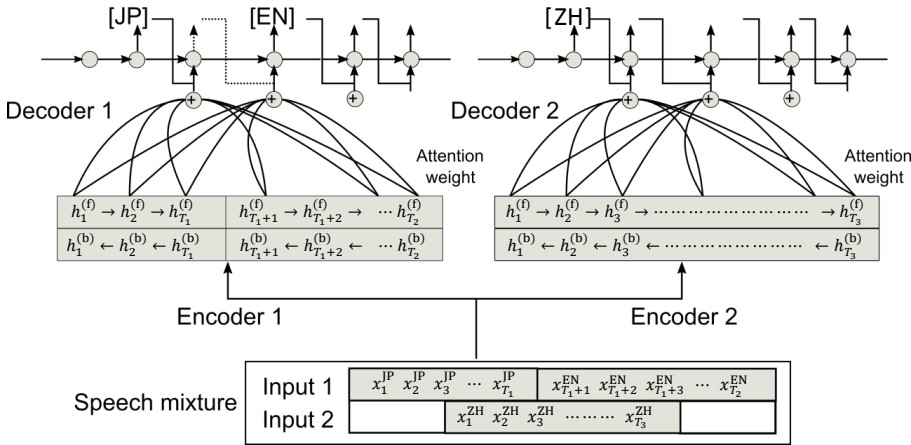

Figure 1: Overview of the proposed end-to-end multi-lingual multi-speaker ASR system. The encoder networks transform an input speech mixture into a set of high-level hidden representations. The decoder network generates output label sequences from each encoder network's output. As a multi-lingual ASR system, the recognizer supports the input of multiple languages and allows speakers to switch languages during an utterance (code-switching). The decoder network generates a language ID followed by a character sequence, and repeats the generation of a language ID and character sequence when the speech includes code-switching.

2016; Settle et al., 2018; Yu et al., 2017). However, Seki et al. (2018a) recently proposed an end-to-end architecture to directly generate multiple hypotheses from a speech mixture without requiring additional auxiliary training signals and separation modules. In addition, Qian et al. (2017) reported that the recognition of two-speaker mixtures using models trained for two-speaker and three-speaker mixtures shows equivalent performance (Qian et al., 2017). Therefore, it may be possible to further eliminate modules which depend on the number of speakers by assuming a large enough maximum number of speakers.

In this paper, we further integrate end-to-end multi-lingual ASR with end-to-end multi-speaker ASR, to propose an unprecedented all-in-one end-to-end multi-lingual multi-speaker ASR system. Figure 1 shows an overview of the multi-lingual multi-speaker ASR system for a mixture of two speakers, where one speaker first speaks in Japanese then in English, while the other speaker speaks in Chinese. In this example, the left side encoder and decoder networks perform recognition of multi-lingual speech with language switching between Japanese and English, and the right side networks perform recognition of the Chinese part. The input speech is a mixture of two speakers and the system is expected to generate hypotheses of two character sequences, one for each speaker. We describe the recognition of multi-lingual speech in Section 2, and the recognition of multi-lingual multi-speaker speech in Section 3, respectively. We evaluate the proposed model using a large dataset of two-speaker mixtures generated by using 10 languages including code-switching in Section 4, and conclude the paper in Section 5.

## 2 MULTI-LINGUAL SPEECH RECOGNITION

### 2.1 END-TO-END ASR NETWORK

We employ a hybrid joint CTC/attention end-to-end ASR framework (Watanabe et al., 2017b). An attention-based encoder-decoder network predicts an $N$-length label sequence $Y = \{y_n \in \mathcal{U} | n = 1, \ldots, N\}$ given a $T$-length input feature vector sequence $O$ and the past label history, where $\mathcal{U}$ denotes a set of character labels. At inference time, the previously emitted labels are used, whereas at training time, the $N$-length reference labels $R = \{r_n \in \mathcal{U} | n = 1, \ldots, N\}$ are used in a *teacher-forcing* fashion. The probability of sequence $Y$ is computed by multiplying the sequence of condi-

tional probabilities of label $y_n$ given the past history $y_{1:n-1}$:

$$p_{\text{att}}(Y|O) = \prod_{n=1}^{N} p_{\text{att}}(y_n|O, y_{1:n-1}). \tag{1}$$

The model is composed of two main sub-modules: an encoder network and a decoder network. The encoder network transforms the input feature vector sequence $O$ into a $C$-dimensional high-level representation $H = \{h_l \in \mathbb{R}^C | l = 1, \ldots, L\}$, where $L$ is the length of the sequence and usually $L < T$ due to subsampling. The decoder network emits labels based on the label history $y$ and a context vector $c$ calculated using an attention mechanism which weights and sums the representation sequence $H$ with attention weight $a$. A hidden state $e$ of the decoder is updated based on the previous state, the previous context vector, and the emitted label. This mechanism is summarized as follows:

$$H = \text{Encoder}(O), \tag{2}$$
$$y_n \sim \text{Decoder}(c_n, y_{n-1}), \tag{3}$$
$$c_n, a_n = \text{Attention}(a_{n-1}, e_n, H), \tag{4}$$
$$e_n = \text{Update}(e_{n-1}, c_{n-1}, y_{n-1}), \tag{5}$$

The hybrid CTC/attention network also includes a connectionist temporal classification (CTC) sub-module, where the probability of label sequence $Y$ is computed as:

$$p_{\text{ctc}}(Y|O) = \sum_Z \prod_{l=1}^{L} p(z_l|z_{l-1}, Y)p(z_l|O), \tag{6}$$

where $p(z_l|z_{l-1}, Y)$ represents state transition, which satisfies monotonic alignment constraints in CTC, and $p(z_l|O)$ is the frame-level label probability computed by

$$p(z_l|O) = \text{Softmax}(\text{Linear}(h_l)), \tag{7}$$

where $\text{Linear}(\cdot)$ is the final linear layer of the CTC sub-module. The summation over all possible sequences $Z \in \mathcal{Z} = \{(z_1, \ldots, z_L)|z_l \in \mathcal{U} \cup \{\texttt{blank}\}, \forall l\}$ is efficiently computed by using the forward-backward algorithm, where $\texttt{blank}$ represents a special label that emits nothing in CTC.

The CTC loss and the attention-based encoder-decoder loss are combined with an interpolation weight $\lambda \in [0, 1]$:

$$\mathcal{L}_{\text{hyb}} = \lambda \mathcal{L}_{\text{ctc}} + (1 - \lambda)\mathcal{L}_{\text{att}}, \tag{8}$$

where we define the CTC and attention losses as:

$$\mathcal{L}_{\text{ctc}} = \text{Loss}_{\text{ctc}}(Y, R) \triangleq -\log p_{\text{ctc}}(Y = R|O), \tag{9}$$
$$\mathcal{L}_{\text{att}} = \text{Loss}_{\text{att}}(Y, R) \triangleq -\log p_{\text{att}}(Y = R|O), \tag{10}$$

based on the cross entropy. Both CTC and encoder-decoder networks are also used in the inference step. The additional CTC objective provides fast and accurate inference during training and decoding thanks to its monotonic alignment property (Watanabe et al., 2017b). In addition, the CTC objective can be used to reduce the computational cost in permutation selection, which will be described in Section 3.2.

## 2.2 AUGMENTED CHARACTER SET

For the recognition of multiple languages, we employ the union of all target language character sets as an augmented character set, i.e., $\mathcal{U} = \mathcal{U}^{\text{EN}} \cup \mathcal{U}^{\text{JP}} \cup \cdots$, where $\mathcal{U}^{\text{EN/JP}/\cdots}$ is a character set of a specific language, as in Watanabe et al. (2017a) and Toshniwal et al. (2018). By using this augmented character set, likelihoods of character sequences can be computed for any language without requiring a separate language identification module. The network is trained to automatically predict the correct character sequence for the target language of each utterance.

## 2.3 AUXILIARY LANGUAGE IDENTIFICATION

Language identification symbols, such as "[EN]" and "[JP]" for English and Japanese, are further added to the augmented character set for an explicit identification of the target language and for modeling the joint distribution of a language ID and a character sequence (Wu et al., 2016; Watanabe et al., 2017a; Li et al., 2018). The language ID is inserted at the beginning of the reference label. The final augmented character set is $\mathcal{U}^{\text{final}} = \mathcal{U} \cup \{[\text{EN}], [\text{JP}], \ldots\}$.

### 2.4 CODE-SWITCHING SPEECH

It is natural for speakers of multiple languages to switch language between or during utterances, a phenomenon known as *code-switching*. Monolingual speakers also frequently use code-switching with foreign named entities and expressions. Code-switching speech is particularly challenging for conventional ASR systems, and typically requires combining multiple mono-lingual systems under a language identification module.

It was showed in Seki et al. (2018b) that multi-lingual speech recognition with code-switching can be more elegantly solved using an end-to-end multi-lingual ASR system trained on a dataset of code-switching speech. Because existing corpora of code-switching speech are limited and thus inadequate for large scale experiments with end-to-end frameworks, Seki et al. (2018b) instead generated a large code-switching corpus by concatenating speech from existing monolingual corpora. Here, we use the same strategy to generate a large dataset of multi-speaker multi-lingual speech with code-switching. We describe the generation procedure in more details in Section 4.1.1.

## 3 MULTI-LINGUAL MULTI-SPEAKER ASR

### 3.1 ATTENTION LOSS FUNCTION FOR END-TO-END MULTI-SPEAKER ASR

When a speech mixture contains speech uttered by $S$ speakers simultaneously, the encoder network generates $S$ hidden representations from the $T$-frame sequence of $D$-dimensional input feature vectors, $O = \{o_t \in \mathbb{R}^D | t = 1, \ldots, T\}$:

$$H^s = \text{Encoder}^s(O), \; s = 1, \ldots, S. \tag{11}$$

In the training stage, the attention-based decoder network uses reference labels $R = \{R^1, \ldots, R^S\}$ for the generation of hypotheses, in a teacher-forcing fashion. There is however here an ambiguity, known as the *permutation problem* (Hershey et al., 2016), as to which reference label should correspond to which estimate. Therefore, the conditional probability of the decoder network for the $u$-th output depends on the selected $v$-th reference label. The probability of the $n$-th label $y_n^{u,v}$ is computed by conditioning on the past reference history $r_{1:n-1}^v$:

$$p_{\text{att}}(Y_{\text{att}}^{u,v}|O) = \prod_n p_{\text{att}}(y_n^{u,v}|H^u, r_{1:n-1}^v). \tag{12}$$

The attention-based decoder network computes the corresponding context vectors, decoder states, and output labels as:

$$c_n^{u,v}, a_n^{u,v} = \text{Attention}(a_{n-1}^{u,v}, e_n^{u,v}, H^u), \tag{13}$$

$$e_n^{u,v} = \text{Update}(e_{n-1}^{u,v}, c_{n-1}^{u,v}, r_{n-1}^{v,v}), \tag{14}$$

$$y_n^{u,v} \sim \text{Decoder}(c_n^{u,v}, r_{n-1}^{u,v}). \tag{15}$$

In the training stage, all possible permutations of the $S$ sequences $R^s = \{r_1^s, \ldots, r_{N_s}^s\}$ of $N_s$ reference labels are considered, and the one leading to minimum loss is adopted for backpropagation, resulting in a permutation-free objective (Hershey et al., 2016; Isik et al., 2016; Yu et al., 2017). Let $\mathcal{P}$ denote the set of permutations on $\{1, \ldots, S\}$. The final attention loss $\mathcal{L}_{\text{att}}$ is defined as

$$\mathcal{L}_{\text{att}} = \min_{\pi \in \mathcal{P}} \sum_{s=1}^{S} \text{Loss}_{\text{att}}(Y_{\text{att}}^{s,\pi(s)}, R^{\pi(s)}), \tag{16}$$

where $\pi(s)$ is the $s$-th element of a permutation $\pi$.

### 3.2 REDUCING THE PERMUTATION SELECTION COST

The final loss of the model is calculated as the weighted sum of the losses from two modules, CTC and encoder-decoder network. As the decoder network takes more computation time than CTC, the permutation of reference labels is selected based on minimizing the CTC loss only: an optimal

permutation $\hat{\pi}$ is calculated based on the CTC network output $Y_{\text{ctc}}^s$ (considered as a random variable) corresponding to $H^s$ and the reference labels, as

$$\hat{\pi} = \arg\min_{\pi \in \mathcal{P}} \sum_{s=1}^{S} \text{Loss}_{\text{ctc}}(Y_{\text{ctc}}^s, R^{\pi(s)}). \qquad (17)$$

This optimal permutation is then used to compute both CTC and attention losses,

$$\mathcal{L}_{\text{ctc}} = \sum_{s=1}^{S} \text{Loss}_{\text{ctc}}(Y_{\text{ctc}}^s, R^{\hat{\pi}(s)}), \qquad (18)$$

$$\mathcal{L}_{\text{att}} = \sum_{s=1}^{S} \text{Loss}_{\text{att}}(Y_{\text{att}}^{s,\hat{\pi}(s)}, R^{\hat{\pi}(s)}), \qquad (19)$$

which are combined as in Eq. 8.

### 3.3 Training a multi-lingual multi-speaker system

By defining the augmented character set as in Section 2.3 and using a multi-lingual multi-speaker corpus, we can train the system to recognize simultaneous speech by multiple speakers in multiple languages. We describe a way to generate such multi-lingual multi-speaker corpus below.

## 4 Experiments

### 4.1 Experimental setup

#### 4.1.1 Corpus

A multi-lingual multi-speaker corpus was generated using the following corpora: WSJ (English) (Consortium, 1994; Garofalo et al., 2007), CSJ (Japanese) (Maekawa et al., 2000), HKUST (Chinese Mandarin) (Liu et al., 2006), and Voxforge (German, Spanish, French, Italian, Dutch, Portuguese, Russian) (VoxForge) for a total of 622.7 hours and 10 languages. The generated mixtures are intended to mimic overlapped speech by two speakers, where each speaker may speak any language and change language during the utterance. Because available corpora typically do not share speakers, we here concatenate utterances in various languages uttered by different speakers. Two such streams are prepared and mixed down into a multi-lingual overlapped speech mixture with code-switching. We now explain this process in more detail. We first sample the number of concatenation $n_{\text{concat}}^1$ and $n_{\text{concat}}^2$ ranging from 1 to $N_{\text{concat}}$ for code-switching within each stream. Then, $n_{\text{concat}}^1$ and $n_{\text{concat}}^2$ utterances are sampled from the union of original corpora. We limit the number of times each utterance can be selected to $n_{\text{reuse}}$, and prevent the same speaker from appearing in both streams to be mixed. The probability of sampling a language is proportional to the duration of its original corpus, while that of sampling an utterance within a language is uniform. Selected utterances are concatenated into respective streams, which are mixed with randomly selected SNR ranging from 0 to $R$ dB. Since the durations of the streams to be mixed are different, we randomize the starting point of the overlapping part by padding the shorter stream with silence. These procedures are repeated until the cumulative duration $d$ of the generated corpus reaches the total duration of the original corpora. In our experiment, $N_{\text{concat}}$ and $n_{\text{reuse}}$ were set to 3, and $R$ was set to 2.5 dB.

#### 4.1.2 Network architecture

We followed the setup of earlier work on joint CTC/attention-based encoder decoder network (Hori et al., 2017). As encoder network, we used a stack of 6-layer VGG network and 8-layer bi-directional long short-term memory (BLSTM) network. For the generation of multiple hypotheses, the encoder network was split at the BLSTM layer: the VGG network generates a single hidden vector, from which two speaker-differentiating 2-layer BLSTMs generate two hidden vectors. The two hidden vectors are further independently fed into the (shared) 6-layer BLSTMs and the decoder network to generate hypotheses for the utterances in the mixture. As input feature, we used 80-dimensional log mel filterbank coefficients with pitch features and their delta and delta delta features extracted

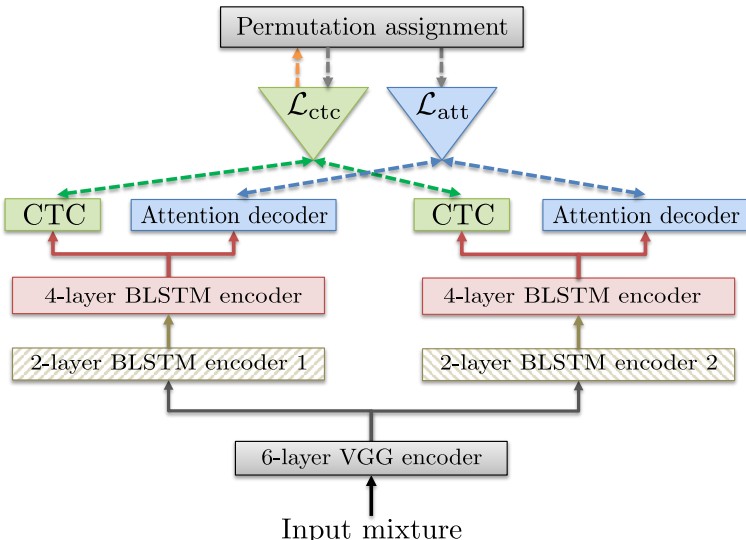

Figure 2: Network architecture with the two speaker mixture case.

using Kaldi (Povey et al., 2011). The BLSTM layer has 320 cells in each layer and direction, and a linear projection layer with 320 units follows each BLSTM layer. The decoder network has a 1-layer LSTM with 320 cells.

### 4.1.3 OPTIMIZATION

We used the AdaDelta algorithm (Zeiler, 2012) with gradient clipping (Pascanu et al., 2013) for optimization. The networks were implemented using ChainerMN (Akiba et al., 2017) and optimized under synchronous data parallelism using 8 GPUs.

In a preliminary experiment, we found that the flat start training of the randomly initialized model using the mixed speech resulted in poor generalization. Therefore, we first trained a randomly initialized network using single-speaker speech without code-switching. The network was then retrained using mixed speech without code-switching, and finally using mixed speech with code-switching. When moving to mixed speech, the other speaker-differentiating encoder was initialized using the already trained one by copying the parameters with random perturbation, $w' = w \times (1 + \text{Uniform}(-0.1, 0.1))$ for each parameter $w$.

### 4.2 RESULTS

### 4.2.1 RECOGNITION PERFORMANCE

Table 1 shows character error rates (CERs) for the generated multi-lingual multi-speaker speech recognized by the baseline multi-lingual single-speaker model. Results are reported separately according to the number of concatenated utterances in each stream within the mixture. We can see that the baseline model has high CERs, over 100%, because the model was trained as a multi-lingual single-speaker ASR system. For the evaluation of the baseline system, the generated hypothesis is duplicated to match the number of references.

Table 2 shows the CERs for the generated speech recognized with the proposed model. Our proposed model significantly reduced the CERs from the baseline model, obtaining an average CER of 44.5%, a 57.5% relative reduction from the baseline.

To investigate the lower bound of CER for the generated corpus, we evaluated the performance of the multi-lingual single-speaker ASR system of Seki et al. (2018b) on each of the multi-lingual streams used in the generated corpus, prior to mixing. This can be considered an oracle result with perfect speech separation. Table 3 shows the oracle CERs. The average CER of the oracle result was 25.4%, showing that there is a room for further performance improvement.

Table 1: Character error rates (CERs %) of mixed speech recognized by the baseline multi-lingual **single-speaker** ASR system.

|  |  | # concat. utt. in softer stream | | | |
|---|---|---|---|---|---|
|  |  | 1 | 2 | 3 | Avg. |
| # concat. utt. | 1 | 107.2 | 107.3 | 109.6 | 108.0 |
| in louder | 2 | 107.5 | 100.5 | 102.0 | 103.3 |
| stream | 3 | 109.1 | 101.1 | 98.1 | 102.7 |
|  | Avg. | 107.9 | 103.0 | 103.2 | 104.7 |

Table 2: CERs (%) of mixed speech recognized by our proposed multi-lingual **multi-speaker** ASR system.

|  |  | # concat. utt. in softer stream | | | |
|---|---|---|---|---|---|
|  |  | 1 | 2 | 3 | Avg. |
| # concat. utt. | 1 | 42.9 | 42.0 | 40.3 | 41.7 |
| in louder | 2 | 41.6 | 46.7 | 47.5 | 45.3 |
| stream | 3 | 40.6 | 47.9 | 50.8 | 46.4 |
|  | Avg. | 41.7 | 45.5 | 46.2 | 44.5 |

Table 3: Oracle CERs (%) of **isolated speech** for each of the utterances appearing in the mixtures used in Tables 1 and 2, recognized by the baseline multi-lingual **single-speaker** ASR system.

|  |  | # concat. utt. in softer stream | | | |
|---|---|---|---|---|---|
|  |  | 1 | 2 | 3 | Avg. |
| # concat. utt. | 1 | 24.4 | 25.3 | 25.3 | 25.0 |
| in louder | 2 | 25.2 | 26.1 | 25.9 | 25.7 |
| stream | 3 | 25.5 | 25.4 | 25.9 | 25.6 |
|  | Avg. | 25.1 | 25.6 | 25.7 | 25.4 |

Table 4: Language ID error rates (LERs %) of the baseline, proposed, and oracle systems.

| baseline | 86.8 |
|---|---|
| proposed | 18.6 |
| oracle | 2.8 |

### 4.2.2 LANGUAGE IDENTIFICATION

Table 4 shows language identification error rates (LERs) of the baseline single-speaker system, our proposed system, and the oracle system described above. The LER was calculated by computing the edit distance between the predicted language IDs and corresponding reference language IDs. Similar to the CERs, there is a gap between the proposed and oracle results, although the obtained LERs were much better than with the baseline single-speaker ASR system.

### 4.2.3 RECOGNITION EXAMPLE

Table 5 shows three examples of transcriptions generated by the proposed model. The first example contains German and Japanese utterances, where there is no code-switching. The results are almost perfect. In the second example, one stream is a concatenation of English speech followed by Chinese speech, and the other is a concatenation of two Japanese utterances. The Japanese result has a few errors but these errors are in fact mostly correct in terms of pronunciation. The third example includes more complex utterances with code-switching, where each stream contains three concatenated utterances. As shown in Table 2, this is the most difficult condition and the CERs are higher than in the other cases. The network did make substitution, insertion, and deletion errors, but there is no swapping of words between sentences, and language IDs are correctly estimated.

## 5 CONCLUSION

We proposed an end-to-end multi-lingual multi-speaker ASR system by integrating a multi-lingual ASR system and a multi-speaker ASR system. The model is able to convert a speech mixture to mul-

Table 5: Examples of multi-lingual multi-speaker recognition result. The CER of hypotheses HYP1 and HYP2 are shown in the parentheses, respectively. Errors are emphasized in red color. "*" is a special token inserted to pad deletion and insertion errors for better readability.

| | |
|---|---|
| Example 1 | REF1: [DE] eine höhere geschwindigkeit ist möglich |
| | HYP1: [DE] eine höh*re geschwindigkeit ist möglich  (CER=2.6%) |
| | REF2: [JP] まずなぜこの内容を選んだかと言うと |
| | HYP2: [JP] まずなぜこの内容を選んだかと言うと  (CER=0.0%) |
| Example 2 | REF1: [EN] grains and soybeans most corn and wheat futures prices were stronger [ZH] 也是的 |
| | HYP1: [EN] grains and soybeans most corn and wheat futures prices were strong*k [ZH] 也是的  (CER=2.8%) |
| | REF2: [JP] えーここで注目すべき**点は例十十一の二重下線部に示すように [JP] アニメですとか |
| | HYP2: [JP] えーここで注目すべきい点は零十十一の二十下線部に示すように [JP] アニメですとか  (CER=8.6%) |
| Example 3 | REF1: [EN] he noted that last week's one hundred eight point dro**p in* **the dow j*on***es industrial average resulted from a slightly weaker dollar [ZH] 呃****子其也蛮普通的 [DE] ich darf nicht |
| | HYP1: [EN] it arter th*e last week's one hundred eight point cround and with* daw jumn the***defter_almove***s resulted from a fl**atly reaker dollar [ZH] 呃是想的其也蛮不同的 [DE] ich darf nicht  (CER= 29.2) |
| | REF2: [ES] sorte*ando los p*r*omontorios de los respaldos los golfos y peníns *ulas formados por las **rodillas [JP] えー次の一手について ま**さまざまな*****議論をしなければいけないというような状況に なっています それでまーえーこれからそれぞれの研究の関連とまーえー このセッションの****見所聞き**所というのを説明したいんですけれども [ZH] 噢 |
| | HYP2: [ES] sortenando los para_el turios de las respa*dos los golfos * penens bulas formados por la* car*ei*das [JP] えー次の一手について まーさまざまなんていうのをしなければいけないとい****うな状況に なっています そいでまーえーこれからそれぞれの研究の関連とまーえー このセッションのみところききとこというのを説明したいんですけれども [ZH] 哦  (CER=20.2) |

tiple hypotheses directly without explicit separation. We evaluated the proposed model using speech mixtures involving two simultaneous speech streams in which the language can switch between 10 languages during the utterance. Our all-in-one multi-lingual multi-speaker system obtained 57.5% relative improvement in CER over the baseline system, and showed strong potential towards this challenging task.

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
