# OpenReview forum: "End-to-End Multi-Lingual Multi-Speaker Speech Recognition"
_ICLR.cc/2019/Conference_

### Official Review · AnonReviewer2 · 2018-11-01
**Interesting but not good enough!**

**Rating:** 3
**Confidence:** 4

**Review:**

This paper presents a framework to train an end-to-end multi-lingual multi-speaker speech recognition system. Overall, the paper is quite clear written.
- Strengthens:
+ Experimental results show consistent improvements in speech recognition performance and language identification performance.

- Weakness:
+ I'm not sure whether the framework is novel. The authors have just mixed training data from several languages to train an end-to-end multi-speaker speech recognition system.
+ I don't see the real motivation why the authors want to make the task harder than needed. The example provided in figure 1 is very rare in reality.
+ The authors claimed that their system can recognise code-switching but actually randomly mixing data from different languages are not code-switching.
+ In general, it would be better to have some more analyses showing what the system can do and why.

---

### Official Review · AnonReviewer1 · 2018-11-02
**Applying known techniques to a non problem**

**Rating:** 3
**Confidence:** 5

**Review:**

The authors propose to build a speech recognition system that has been trained to recognize a recording that has been produced by mixing multiple recordings from different languages together, and allowing for some code switching (also done artificially by concatenating different recordings).

While this sounds fancy and like a hard problem, it is in fact easier than recognizing two speakers that have been mixed together speaking the same language, which has already been solved in (Seki, 2018a), from what I can tell. I don't see any contribution in this paper, other than explaining how to create an artificial (un-realistic) database of mixed speech in multiple languages, and then training a multi-speaker end-to-end speech recognition system on that database.

---

### Official Review · AnonReviewer3 · 2018-11-07
**not enough novelty**

**Rating:** 3
**Confidence:** 4

**Review:**

This paper presents an end-to-end system that can recognize single-channel multiple-speaker speech with multiple languages.

Pros:
- The paper is well written.
- It shows the existing end-to-end multi-lingual ASR (Seki et al., 2018b) and end-to-end multi-speaker ASR (Seki et al., 2018a) techniques can be combined without any change to achieve reasonable performance.
- It demonstrates the challenge of single-channel multi-lingual multiple-speaker speech recognition, and compares the performance of the multiple-speaker system on the mixed speech and the single-speaker system on the isolated speech.

Cons:
- It lacks novelty: the proposed framework just simply combines the two existing techniques as mentioned above.
- The training and evaluation data are both artificially created by randomly concatenating utterances with different languages from different speakers with different context. I am not sure of how useful the evaluation is, since this situation is not realistic. Also, currently it cannot test the real code-switching since the utterances are not related and not from the same speaker.
- There are not enough analyses. E.g. it would be good to analyze what contributes to the gap between the single-speaker ASR system performance on the isolated speech and the multi-lingual multi-speaker ASR system on the mixed speech. How well does the proposed end-to-end framework perform compared to a two-step framework with speaker separation followed by multi-lingual single-speaker ASR?

---

### Meta-Review · Area_Chair1 · 2018-12-14
**Limited novelty**

**Confidence:** 5
**Recommendation:** Reject

**Metareview:**

The authors present a system for end-to-end multi-lingual and multi-speaker speech recognition. The presented method is based on multiple prior works that propose end-to-end models for multi-lingual ASR and multi-speaker ASR; the work combines these techniques and shows that a single system can do both with minimal changes.

The main critique from the reviewers is that the paper lacks novelty. It builds heavily on existing work, and  does not make any enough contributions to be accepted at ICLR. Furthermore, training and evaluations are all on simulated test sets that are not very realistic. So it is unclear how well the techniques would generalize to real use-cases. For these reasons, the recommendation is to reject the paper.